# Fast Image Encryption Algorithm for Logistics-Sine-Cosine Mapping

**DOI:** 10.3390/s22249929

**Published:** 2022-12-16

**Authors:** Pengfei Wang, Yixu Wang, Jiafu Xiang, Xiaoling Xiao

**Affiliations:** School of Computer Science, Yangtze University, Jingzhou 434023, China

**Keywords:** color image, logistics-sine-cosine, substitution, diffusion, image encryption

## Abstract

Because images are vulnerable to external attacks in the process of network transmission and traditional image encryption algorithms have limitations such as long encryption time, insufficient entropy or poor diffusion of cipher image information when encrypting color images, a fast image encryption algorithm based on logistics-sine-cosine mapping is proposed. The algorithm first generates five sets of encrypted sequences from the logistics-sine-cosine mapping, then uses the order of the encryption sequence to scramble the image pixels and designs a new pixel diffusion network to further improve the key sensitivity and plain-image sensitivity of the encryption algorithm. Finally, in a series of security analysis experiments, the experimental image Lena was tested 100 times, and the average encryption time was 0.479 s. The average value of the information entropy, pixel change rate and uniform average change intensity of the cipher image reached 7.9994, 99.62% and 33.48%, respectively. The experimental results show that the fast image encryption algorithm based on logistics-sine-cosine mapping takes less time to encrypt, and the cipher image has good information entropy and diffusivity. It is a safe and effective fast image encryption algorithm.

## 1. Introduction

Traditional image encryption, such as IDEA, AES, DES and 3-DES [1], is mainly aimed at data stream encryption, which is not only inefficient and computationally expensive, but these limitations become more and more obvious when encrypting color images. Existing image encryption is roughly divided into spatial domain-based pixel scrambling, chaotic system-based, transformation-domain-based, secret segmentation and secret sharing, neural network and cellular automata-based image encryption.

An image encryption algorithm based on spatial-domain pixel scrambling refers to the use of a specific algorithm to change the position of 3-channel pixels in an image without changing the RGB value of the pixel. As a result, such algorithms are generally less secure, but also relatively low in computational complexity.

An image encryption method based on chaos [2,3,4,5,6] uses the self-similarity of chaos to generate a random sequence as a factor sequence of encryption. The key space is large, the encryption speed is fast, but its limitations are also obvious: there is a trivial key. At present, only high-dimensional chaos or super chaos sacrifice speed can break this limitation.

The image encryption algorithm based on the transformation domain transforms the image from the airspace domain to the frequency domain encryption and then transforms it into the airspace, but because of the limitation of computer accuracy, part of the data accuracy will be lost during the encryption process, which is lossy encryption.

Based on secret segmentation and secret sharing image encryption, the image is encrypted into m cipher images and a threshold k (m ≥ k) is set; only by obtaining n (n ≥ k) different cipher images can the plain image be restored, and its advantage is that even if part of the data is lost, the plain image can still be restored, but the data redundancy is relatively large.

Image encryption based on neural networks and cellular automata has a large key space and good noise resistance because of the nonlinear or self-organizing nature of the encryption system. In addition, there are image encryption algorithms based on DNA [7,8,9,10,11], RNA [12,13,14,15,16], image selectivity [17], optical encryption system [18,19,20,21,22] and random grid [23,24,25,26,27] visualization.

### Motivations and Contributions

Because images are vulnerable to external attacks in the process of network transmission and traditional image encryption algorithms have limitations such as long encryption time, insufficient entropy or poor diffusion of cipher image information when encrypting color images, a fast image encryption algorithm based on logistics-sine-cosine mapping is proposed that generates encryption sequences with high efficiency based on logistics-sine-cosine mapping and designs an efficient pixel scrambling method and three-round pixel diffusion based on the encryption sequence, which has high encryption efficiency, sufficient information entropy and good diffusion of cipher images.

## 2. Related Work

In terms of the research status of color image encryption: Liu et al. [7] proposed a color image hyperchaotic encryption algorithm using DNA dynamic coding and adaptive arrangement and designed a new four-dimensional hyperchaotic system and plain-image-related adaptive permutation method, which makes the operation result more unpredictable and improves the sensitivity of the algorithm to plain images and keys. Zou et al. [8] used DNA coding and sequencing to construct short DNA strands and long DNA strands, in which short DNA strands were used for DNA strand exchange and long DNA strands were used for DNA strand diffusion, and the experimental results verified that the proposed algorithm has high safety and excellent efficiency. Samiullah et al. [9] propose a new symmetric block cipher scheme. It uses DNA-based dynamic S-boxes and hyperchaotic systems connected with MD5 to generate obfuscation and diffusion to encrypt color images, greatly improving the computational efficiency of encryption algorithms. Liu et al. [10] proposed a hyperchaotic image encryption algorithm based on plain image information and DNA calculation, which enhanced the relationship between pixel position, grayscale value and plain image, which not only showed excellent encryption performance, but also resisted various typical attacks. Liang et al. [11] introduced DNA strand displacement and a four-dimensional multi-stable superchaotic system, which increased the connection between the original image and the key so that the encryption algorithm has good resistance to exhaustive attacks, statistical attacks and known original image attacks on RGB color images. Chu et al. [12] proposed a three-dimensional image encryption method based on the memristor chaotic system and RNA cross-mutation. During the entire encryption process, the Arnold matrix, RNA encoding and decoding rules, and crossover and mutation algorithms are determined by the memristor chaotic system, which can encrypt 3D images at the same time and effectively resist various attacks. Lu et al. [13] proposed an adaptive compression encryption method for content based on CS and ribonucleic acid (RNA). Chaotic sequences driven by pure image hashes are utilized throughout the encryption process, enhancing the correlation between the algorithm and the pure image, improving resistance to known and selected plain image attacks. Ghorbani et al. [14] proposed a safe and effective encryption scheme based on ribonucleic acid (RNA) and 2D He and non-maps, which improved the advantages of the most advanced algorithm. Sha et al. [15] proposed an image cryptographic system based on the genetic center law (GCD), Gartner–Moria–Pratt (KMP) algorithm and chaotic system, which realizes DNA-level bidirectional pixel shuffling through shared stack push operation to accelerate the shuffling of overall pixels and replaces pixel values by simulating the protein synthesis process in GCD, with satisfactory low time complexity and excellent safety effects. Zhang et al. [16] defined the 3D bit plane based on the genetic center method, converted k raw images into 8-bit binary and converted to 3D matrix, arranged the 3D matrix by rotating the bit plane and arranged between the bit planes, encoded the scrambled 3D matrix into DNA code and obtained encrypted images through RNA decoding operations, which has strong security and ideal performance. Hasimoto-Beltran et al. [17] propose a new integer chaos-based coupling mapping lattice CML, and encrypt images based on random bit flipping of bitstreams that are not exposed to the attacker’s dynamic reference point (DRP) and random selection of the CML byte trajectories of DRP and bit flipping processes, which has good security and scalability. Li et al. [18] studied an optical image encryption scheme based on fractional Fourier transform and a five-dimensional host-induced nonlinear fractional-order laser ultra-chaotic system and implemented the system by analyzing the dynamic characteristics of the proposed fractional-order laser super chaotic system and using the DSP platform. An image encryption scheme combining BP neural network, GF(17) domain diffusion and hyperchaotic random point scrambling algorithm is proposed, which provides new research prospects for optical image encryption. Kumar et al. [19] proposed image encryption of light carrying optical vortex arrays to minimize the quality degradation during actual implementation, providing an effective method for designing optical cryptographic systems. Chen et al. [20] proposed an exciting optical image encryption method based on spatial nonlinear optics, which realized the high degree of freedom to explore and apply light and provided new research prospects for the development of optical encryption for the protection of information in various optical structures and optical materials. Cheremkhin et al. [21] proposed the optical implementation of an information optical encryption system using a new multi-level customizable digital data container with high data density, which has efficient error correction ability. Kumar et al. [22] propose two-dimensional XORor-logic manipulation of beams carrying optical vortex arrays, which brings new possibilities for high-dimensional data processing, cryptography, and computing. Chen et al. [23] propose an RG-based VSS scheme that encodes multiple secret images at once, which has no pixel expansion and higher secret sharing ability than traditional VC-based VSS. Chen et al. [24] proposed a new RG-based VSS scheme, which not only improves the capacity of secret communication, but also avoids the problem of pixel expansion, thereby significantly reducing the overhead of storage and communication. De et al. [25] improved several schemes and provided many upper limits for the random grid model by exploiting (many) known outcomes in the deterministic model and provided new schemes for the deterministic model by utilizing some of the known outcomes of the random grid model. Lin et al. [26] propose a random grid-chain-based encryption scheme suitable for processing batch binary, grayscale, and color images, and its decryption process is done by the human vision system, which requires neither additional pixel expansion nor any coding base matrix. Shyu et al. [27] devised an innovative algorithm for visual multi-secret sharing using circular or cylindrical random meshes, which is able to share multiple (instead of just or two) secret images in two shares without causing any additional pixel expansion compared to traditional visual cryptography. Cheng et al. [28] adopt ultra-chaotic system and permutation-diffusion architecture to realize the encryption of color images by mixing RGB components, strengthen the dependence between the three components of RGB, and greatly improve the algorithm’s ability to resist statistical attacks and differential attacks. Xiong et al. [29] proposed a color image chaotic encryption algorithm combining cyclic redundancy check (CRC) and nine-house graph, after moving and shuffling pixels through the nine-house graph theory, extracting R, G, and B components to form a binary matrix, and finally cyclic shifting according to cyclic redundancy check (CRC), which better solved the problem of insufficient entropy of cipher image information. Chai et al. [30] used dynamic DNA encryption and chaos system to encrypt color images, first shuffling the three components R, G and B by using plain synchronous components and then converting the permutations and combinations into DNA matrix for encryption according to DNA coding rules, which effectively improved the ability of the algorithm to resist plain attacks. Tsafack et al. [31] proposed a new two-dimensional trigonometric logistics-sine-cosine mapping, which first generates three sets of key streams from the map and calculates the Hamming distance in combination with the image R, G and B components, and finally XOR the output distance vector with the key stream to realize the encryption of the image, which greatly improves the security of the cryptosystem. Fei et al. [32] proposed an image encryption algorithm for jump diffusion based on a two-dimensional absolute sine–cosine coupling chaotic system (2D-ASCC) with higher complexity and better pseudo-randomness, which effectively improves the security performance of the encryption algorithm. Hoang et al. [33] proposed a chaos-based multi-image encryption algorithm, which adopts the permutation diffusion architecture for the first time and constructs the disturbance amount according to the pixel coordinates and the original image content, respectively, in the process of pixel arrangement and diffusion, so that the encryption algorithm has good resistance to differential attacks. Sridevi et al. [34] proposed a chaos-assisted color image encryption algorithm. First, the color image is split into RGB planes, and then chaotic maps and attractors are introduced to double obfuscate and diffuse the images, and finally, the separated RGB planes are merged to produce encrypted images, which greatly improves the anti-attack ability of the algorithm. Pour et al. [35] proposed an image encryption algorithm with a large key space and good attack resistance. The algorithm uses chaos game representation (CGR) images to encrypt images, converting image pixels into binary and combining them to generate binary strings, with each pair of bits acting as a letter of the DNA sequence. Zhang et al. [36] designed a two-dimensional sine–cosine coupled chaotic system (2D-SCCM) with better randomness and traversability and wider hyperchaotic range, and proposed a color image encryption algorithm with high encryption efficiency and strong security based on the chaotic system. The algorithm first uses a combination of pure image and hash function to generate a key, then fuses the random sequence generated by 2D-SCCM and Arnold map to construct an S-Box, and finally encrypts the color image by constructing an S-Box, chaos system and hash function. Su et al. [37] proposed an image encryption algorithm based on binary tree space–time chaos and middle-order traversal. The algorithm uses the intermediate traversal sequence to sort the plain images and introduces a coupled mapping lattice to generate a chaotic sequence; it sets the chaotic interference value and finally performs an XOR operation on the adjacent pixels of the image to encrypt the image, which has good robustness. Zhang et al. [38] proposed a multi-digital image encryption algorithm with excellent image encryption effect and security. The cyclic matrix is constructed by logical mapping, and the sparse transformation of plain images is realized by combining discrete cosine transformation (DCT), and the Lorenz chaotic system and logic graph are used to generate chaotic bond sequences to encrypt images. Li et al. [39] proposed a multi-image encryption scheme based on a DNA chaos algorithm based on a computational integral imaging framework. By merging multiple images into one image with a computational integral imaging algorithm, the efficiency and security of image encryption are significantly improved, and the contour effect caused by the DNA encryption algorithm is solved because of the high randomness of the chaotic algorithm. To reduce redundant encryption operations and focus on protecting important areas, Song et al. [40] proposed a fast image encryption algorithm based on the object detection model and chaotic system, so that the detected objects were well-protected. Yu et al. [41] proposed an image encryption algorithm with good attack resistance. The algorithm uses the scattering medium as the physical key and fuses the four-step phase-shifted digital hologram to convert the plain image into four noise-like holograms, and achieve the effect of encrypting the image by combining the wavelet transform (LWT) into a new image. Adeel et al. [42] proposed a new method for encrypting color images based on a set of integer keys and a one-dimensional chaotic system, which improves the security of color image encryption by selecting the key associated with the chaotic system to optimize the security of the cryptographic image, which can be used for the secure encryption of image data. Basha et al. [43] proposed a color image encryption technology based on bit-level chaos, which decomposes the color image into red, blue and green color image components based on the logistic-sine-tent-Chebyshev (LSTC) map for encryption, which has good resistance to statistical attacks, differential attacks and brute force attacks. Hosny et al. [44] proposed a new color image cryptographic system that changes the value of image pixels according to the 2D logical sine diagram to reconstruct the scrambled image, and uses the key to diffuse the scrambled image to obtain an encrypted image, which has the characteristics of wide key space, high key sensitivity and good encryption effect. Wang et al. [45] proposed a color image encryption algorithm based on Fisher-Yates scrambling and DNA subsequence operations (elongation operation, truncation operation, deletion operation and insertion operation), which uses the chaotic sequence generated by the Chen system and the Fisher-Yates scrambling method to scramble the plain image of the R, G and B channels, and then introduces DNA coding rules to destroy the scrambled plain-image information to obtain encrypted color images, which have good performance and can resist various typical attacks. Based on logistic sine–cosine mapping, the encryption algorithm in this paper generates the encryption sequence with high efficiency, and an efficient pixel scrambling method and three-round pixel diffusion method are designed based on the encryption sequence, which has high encryption efficiency, sufficient information entropy and good diffusion of cipher images. The encryption diagram and decryption diagram are shown in Figure 1 and Figure 2, respectively.

## 3. Methods

### 3.1. Logistics-Sine-Cosine Mapping

The logistics-sine-cosine mapping [31] is a map composed of two seed maps of logistics and sine maps cascading with cosine maps:(1)$$x_{i+1}=\cos(\pi (4rx_{i}(1-x_{i})+r\sin(\pi x_{i})-0.5))$$

The mapped bifurcation plot, the Lyapunov index, is shown in Figure 3.

As long as *r*∈[0, 1], *x*∈(0, 1), the map can exhibit complex and random behavior.

### 3.2. Fast Image Encryption Algorithm for Logistics-Sine-Cosine Mapping

#### 3.2.1. Key Distribution and Cryptographic Algorithm Structure

Considering the accuracy of computer calculations, the key is a 320-bit binary number generated by a random binary number generator, consisting of 32-bit binary numbers $x_{0},x_{1},x_{2},x_{3},x_{4}$ and $r_{0},r_{1},r_{2},r_{3},r_{4}$, where *x* is the initial value and *r* is the control parameter.

The initial value *x* and the control parameter *r* are converted from Equation (2) to floating-point numbers:(2)$$\{{}_{r_{float}=\sum_{i=1}^{32}r_{bin}2^{-i}}^{x_{{}_{float}}=\sum_{i=1}^{32}x_{{}_{bin}}2^{-i}}$$
(3)$$F(\omega ,\varphi ,c,primary_{i})=primary_{i}\cdot \sin(\omega \cdot primary_{i}+\varphi )+c$$
(4)$$primary_{i}=primary_{0}+i\cdot step$$

Bring each set of *x*, *r* into the logistics-sine-cosine map separately to generate a set of chaotic sequences ω, φ, *c*, ${primar}_{0}$, step of length 5. Then bring each set of ω, φ, *c*, ${primar}_{0}$, step into the third and fourth iterations to generate an encryption sequence with length *H* × *W* × 3; finally, five sets of encryption sequences, $S_{0}$, $S_{1}$, $S_{2}$, $S_{3}$ and $S_{4},$ are obtained.

#### 3.2.2. Pixel Scrambling

Assuming that the plain image *P* is a color image of *H* × *W*, the five sets of encryption sequences with length *H* × *W* × 3 generated by the key are $S_{0}$, $S_{1}$, $S_{2}$, $S_{3}$ and $S_{4}$. The rules for scrambling are as follows:

Assuming that the positive sequence corresponding to the encryption sequence $S_{0}$ is $S_{sorted}$ (sorted from smallest to largest), there must be $S_{sorted}\overset{X_{scramble}}{\to }S_{0}$, where $X_{scramble}$ is the scramble operation.

Here is how to scramble an image:

Step 1: use sequence $X_{scramble}$ to store the position information of the elements in the positive sequence $S_{sorted}$ in the $S_{0}$ of the encrypted sequence; for example, the Ath element B in the positive sequence $S_{sorted}$ is the *C*th element in the encryption sequence $S_{0}$; then the value of the *A*th element of sequence $X_{scramble}$ is *C*.

Step 2: transform the image *P* into a sequence $S_{P}$ with a one-dimensional length *H* × *W* × 3, so that $S_{P}$ = $(S_{P}$ + $S_{0})mod256$ to get a new $S_{P}$; use sequence $X_{scramble}$ to scramble the sequence $S_{P}$; for example, the *D*th element value in the sequence $S_{P}$ is *E* and the *D*th element value in sequence $X_{scramble}$ is *F*; then the *F*th element value in the scrambled sequence $S_{SP}$ is *E* ($S_{SP}$ is the sequence after scrambling).

For the sake of description, see the following example:

Assuming that the image *P* is a color image of 2 × 2, the generated encryption sequence $S_{0}$ = (2.1, 2.8, 2.6, 2.7, 1.3, 1.6, 0.55, 0.53, 0.74, 1.8, 1.2, 2.5); then the corresponding positive sequence $S_{sorted}$ = (0.53, 0.55, 0.74, 1.2, 1.3, 1.6, 1.8, 2.1, 2.5, 2.6, 3.7, 2.8); it is easy to conclude that the position of the 0th element in $S_{sorted}$ in the $S_{0}$ is 7th (counting from 0); then the value of the 0th element of sequence $X_{scramble}$ is 7 and so on to get the complete sequence $X_{scramble}$ = (7, 6, 8, 10, 4, 5, 9, 0, 11, 2, 3, 1). The image *P* is transformed into a sequence $S_{P}$ with a one-dimensional length of 2 × 2 × 3, so that $S_{P}$ = $(S_{P}$ + $S_{0})mod256$ to get a new $S_{P}$, and then sequence $X_{scramble}$ is used to scramble the sequence $S_{P}$; for example, the value of the 0th element in the sequence $S_{P}$ is 111, and the value of the 0th element in the sequence $X_{scramble}$ is 7, and then the value of the 7th element in the scrambled sequence $S_{SP}$ is 111. Figure 4 shows a schematic diagram of the position change before and after the *P*-pixel of the image is scrambled.

When decrypting, perform the reverse operation of pixel scrambling:

Step 1 uses sequence $X_{sort}$ to store the position information of the element in the $S_{0}$ of the encryption sequence in the positive sequence $S_{sorted}$; for example, the *C*th element *B* in the encryption sequence $S_{0}$ is the *A*th element in the positive sequence $S_{sorted}$, and then the value of the *C*th element of the sequence $X_{sort}$ is *A*.

Step 2: suppose the image after encryption scrambling is $P_{s}$. The $P_{s}$ is reduced to a one-dimensional sequence $S_{SP}$ of length *H* × *W* × 3, and the sequence $S_{SP}$ is restored by using sequence $X_{sort}$; for example, the *F*th element value in the sequence $S_{SP}$ is *E*, and the *F*th element value in sequence $X_{sort}$ is *D*, and then the *D*th element value in the restored sequence $S_{P}$ is *E*. Finally, let $S_{P}$ = $(S_{P}$ − $S_{0})mod256$ to restore the pixel scrambling.

#### 3.2.3. Pixel Diffusion

Suppose that the image $P_{S}$ before diffusion is an *H* × *W* color image after scrambling; first, convert the image $P_{S}$ into a one-dimensional sequence $S_{SP}$ of length *H* × *W* × 3, and then use the $S_{1}$ to scramble the $S_{2}$, $S_{3}$ and $S_{4}$. The scrambling rules are the same as the pixel scrambling rules to obtain a new $S_{2}$, $S_{3}$ and $S_{4}$ sequence.

For ease of description, here the functions *F*(*L*, *n*) and *G*(*M*, *n*) are set, and *F*(*L*, *n*) represents shifting *L* left by *n* units; where *L* is a sequence, *n* is an integer, and when *n* is negative, *L* is shifted right by -*n* units. For example, *L* = (0,1,2,3,4,5,6,7,8,9), *n* = 3 and then *F*(*L*, *n*) = (3,4,5,6,7,8,9,0,1,2) for the function *G*(*M*, *n*); assuming that the binary number corresponding to *M* is $M_{b}$, then *G*(*M*, *n*) means that the *n* bits of the $M_{b}$ low (right) are shifted out, and the other numbers are shifted *n* bits to the right, and the decimal number corresponding to the zero is filled in the *n* empty bits of the high (left); for example, *G*(5, 1) = 2.

When *X* is the main diffusion sequence, $S_{2}$ is the round of the encryption sequence; when *Y* is the main diffusion sequence, $S_{3}$ is the round of the encryption sequence, and when *Z* is the main diffusion sequence, $S_{4}$ is the round of the encryption sequence. Given the space, only *X* is given here as the diffusion rule for the principal diffusion vector:

Step 1 divides the sequence $S_{SP}$ into three groups of *X*, *Y* and *Z* so that *N* = *H* × *W*, *X* = $S_{SP}$ [0:*N*], *Y* = $S_{SP}$ [*N*:2 × *N*], *Z* = $S_{SP}$ [2 × *N*:3 × *N*] and *round* = $\lfloor \log_{2}^{N}\rfloor +1$; divide the encryption sequence $S_{2}$ into three groups of sequences, $K_{x}$, $K_{y}$ and $K_{z}$; let $K_{x}$ = $S_{2}$ [0:*N*], $K_{y}$ = $S_{2}$ [*N*:2 × *N*] and $K_{z}$ = $S_{2}$ [2 × *N*:3 × *N*].

Step 2: set *X* as the main diffusion sequence, and the other two as the secondary diffusion sequence; that is, $V_{p}$ is *X*, $V_{sp1}$ is *Y* and $V_{sp2}$ is *Z*, and the diffusion steps are as follows:$$\begin{array}{l} V_{SP1}=\mathrm{mod}(V_{SP1}+V_{P}+K_{V_{P}},256)\\ V_{P}=\mathrm{mod}(V_{P}+V_{SP1}+K_{V_{SP1}},256)\\ V_{SP2}=\mathrm{mod}(V_{SP2}+V_{SP1}+K_{V_{SP1}},256)\\ V_{SP2}=\mathrm{mod}(V_{SP2}+V_{P}+K_{V_{P}},256)\\ V_{P}=\mathrm{mod}(V_{P}+V_{SP2}+K_{V_{SP2}},256)\\ V_{SP1}=\mathrm{mod}(V_{SP1}+V_{SP2}+K_{V_{SP2}},256)\\ L\mathrm{ast}=0,n=1\\ V_{P}=F(V_{P},Last)\\ New=G(N,n)\\ V_{P}=F(V_{P},New)\\ n=n+1\\ Last=-New\\ V_{SP1}=\mathrm{mod}(V_{SP1}+V_{P}+K_{V_{P}},256)\\ V_{P}=\mathrm{mod}(V_{P}+V_{SP1}+K_{V_{SP1}},256)\\ V_{SP2}=\mathrm{mod}(V_{SP2}+V_{SP1}+K_{V_{SP1}},256)\\ V_{SP2}=\mathrm{mod}(V_{SP2}+V_{P}+K_{V_{P}},256)\\ V_{P}=\mathrm{mod}(V_{P}+V_{SP2}+K_{V_{SP2}},256)\\ V_{SP1}=\mathrm{mod}(V_{SP1}+V_{SP2}+K_{V_{SP2}},256) \end{array}$$

Repeat Lines 8 through 18 of the above equation until *n* > *round*, and finally splice and recombine the three groups of sequences of *X*, *Y* and *Z* into sequences $S_{DP}$, which can complete a diffusion.

To illustrate the above steps more vividly, assuming that *N* = 10, *round* = $\lfloor \log_{2}^{N}\rfloor +1$ = 4, the change element is $y_{0}$ and Figure 5 is a diffusion diagram of the *Y* main diffusion sequence, it can be seen that when the $y_{0}$ changes, after *round* + 1 small diffusion, it will lead to the change of all elements.

In general, the diffusion process is that the image data is divided into three groups, three rounds of diffusion; each round of diffusion selects one of *X*, *Y* or *Z* as the main diffusion sequence and the other two as the secondary diffusion sequence. The main diffusion vector diffuses at the same time to carry out round cycle shift, and finally realizes that the change of any element will cause all the output elements to change, which further improves the key sensitivity and plain sensitivity of the encryption algorithm in this paper.

When decrypting, the reverse operation of the encryption operation on the cipher image can be performed, limited by space, and only *X* is given as the corresponding decryption step when the main diffusion sequence:$$\begin{array}{l} n=round\\ New=G(N,n)\\ X=F(X,New)\\ Y=\mathrm{mod}(Y-Z-K_{Z},256)\\ X=\mathrm{mod}(X-Z-K_{Z},256)\\ Z=\mathrm{mod}(Z-X-K_{X},256)\\ Z=\mathrm{mod}(Z-Y-K_{Y},256)\\ X=\mathrm{mod}(X-Y-K_{Y},256)\\ Y=\mathrm{mod}(Y-X-K_{X},256)\\ X=F(X,-New)\\ n=n-1\\ \mathrm{Repeat}lines\;2-11\;\mathrm{until}\;n>round\\ Y=\mathrm{mod}(Y-Z-K_{Z},256)\\ X=\mathrm{mod}(X-Z-K_{Z},256)\\ Z=\mathrm{mod}(Z-X-K_{X},256)\\ Z=\mathrm{mod}(Z-Y-K_{Y},256)\\ X=\mathrm{mod}(X-Y-K_{Y},256)\\ Y=\mathrm{mod}(Y-X-K_{X},256) \end{array}$$

## 4. Experimental and Safety Analysis

The experimental environment in this article is as follows: the hardware environment is Intel(R) Core(TM) i5-7200U CPU @ 2.50 GHz, 8.00 GB RAM. The software environment is Windows 10, Python 3.8.8.

Experimental data (Figure 6): to facilitate the comparison of experimental results, color images 1024 × 1024 2.2.01, 512 × 512 Lena and special images (256 × 256 pure black images) in the public image dataset USC-SIPI “Aerials” were selected for related safety-test experiments.

### 4.1. Key Randomness Analysis

The key for the encryption algorithm in this article is a 320-bit binary number generated by the random binary number generator. Therefore, 1000 keys are generated by random binary number generator, and the NIST statistical test suite is used to test the randomness of the generated keys, and the results are averaged to evaluate the randomness of the encryption algorithm keys in this paper. The test results are shown in Table 1, and in the 15 tests of the NIST statistical test, the randomly generated binary numbers all passed the test, indicating that the 320-bit binary numbers generated by the random binary number generator in this paper are of good randomness and can be used for image encryption.

### 4.2. Key Security Analysis

The key space of the encryption algorithm in this article is 2^320^ > 2^100^, which can theoretically resist any form of brute force attack [46], and the following is further proof of the actual size of the key space.

In this paper, the number of bit change rate (*NBCR*) [47] is used to evaluate the key sensitivity of the algorithm.
(5)$$NBCR\;(P_{1},P_{2})=\frac{Ham\;(P_{1},P_{2})}{3*H*W*8}$$

That is the ratio of the total Hamming distance to the total number of bits.

Randomly generate a 320-bit binary number key, change one bit on the basis of the original key from high bit to low bit, encrypt the plain image *P* with the key before and after the change, obtain two cipher images of $C_{1}$ and $C_{2}$, and calculate the *NBCR* ($C_{1}$, $C_{2}$) of the two images of $C_{1}$ and $C_{2}$, and then decrypt the cipher image with the key before and after the change $C_{1}$ to obtain the images $D_{1}$ and $D_{2}$, and calculate the *NBCR* ($D_{1}$, $D_{2}$). The results are shown in Figure 7. *NBCR* ($C_{1}$, $C_{2}$) and *NBCR* ($D_{1}$, $D_{2}$) are both around 0.5 and the oscillation amplitude is extremely small, indicating that the key sensitivity of the algorithm in this paper is good, and changing any bit in the key will lead to a completely different cipher image and decryption image.

### 4.3. Image Encryption/Decryption Experiments

The experimental images shown in Figure 6 were encrypted 100 times, and the key was randomly generated each time, as shown in Figure 8. The results of a randomly selected experiment, algorithm time complexity and the average running time are shown in Table 2 (*W* and *H* represent the width and height of the encrypted image, respectively). It can be seen that the encryption algorithm in this paper is lossless, and the encryption time is shorter than that of the literature [42,43,44,45]. Although the proposed encryption algorithm is slightly better than the literature [42,43,44,45] in the time comparison experiment of encrypting Lena images, it can be seen that the encryption time of the proposed algorithm increases with the size of the encrypted image, and the encryption efficiency of the proposed algorithm will gradually decrease as the image size increases, and further improvement is needed.

### 4.4. Statistical Analysis

#### 4.4.1. Histogram Analysis

The image histogram visually reflects the distribution of gray values of the image, and an attacker can use this feature to analyze the statistical law between plain and cipher images to potentially decipher the key. For this to happen, the histogram of the cipher image must be evenly distributed and completely different from the histogram of the plain image, and the more evenly distributed, the less statistics an attacker will be able to obtain. The pixel histogram of the cipher image based on the encryption scheme in this paper is shown in Figure 9, and the pixel histogram of the cipher image is completely different from the original image and evenly distributed.

#### 4.4.2. Correlation Analysis

In an image, neighboring pixels are often highly correlated, and if this high correlation is not eliminated, an attacker can use this feature to predict its surrounding pixels from one pixel and eventually restore the entire plain image, so it is necessary to completely avoid statistical analysis attacks, eliminating these strong correlations. The correlation coefficient for each upward direction is calculated as follows (horizontal, vertical and diagonal):
(6)$$R_{xy}=\frac{\mathrm{cov}(x,y)}{\sqrt{D(x)}\sqrt{D(y)}}$$
(7)$$E(x)=\frac{1}{N}\sum_{i=1}^{N}x_{i}$$
(8)$$D(x)=\frac{1}{N}\sum_{i=1}^{N}{\left(x_{i}-E(x)\right)}^{2}$$
(9)$$\mathrm{cov}(x,y)=\frac{1}{N}\sum_{i=1}^{N}\left(x_{i}-E(x))(y_{i}-E(y)\right)$$
where *x* and *y* are two adjacent pixels, *N* is the total number of pixels in the image, $R_{xy}$ is the correlation coefficient of *x*, *y*, cov(*x*,*y*) is the covariance of *x* and *y*, $\sqrt{D\left(x\right)}$ is the standard deviation, *D*(*x*) is the variance and *E*(*x*) is the mean.

The experimental images were encrypted 100 times; each time the key was randomly generated and 3000 pairs of neighboring pixels were randomly selected. The correlations before and after encryption are shown in Figure 10, and the correlation coefficients are shown in Table 3. Most of the correlation coefficients after encryption are close to 0, indicating that the confusion and diffusion of the encryption scheme in this paper are good, which further reduces the occurrence of statistical analysis attacks, but some correlation coefficients are large, and further work is needed to optimize them.

#### 4.4.3. Information Entropy Analysis

Information entropy is a quantitative measure of how random a signal source is. That is, information entropy can be used to measure the randomness of an image, which calculates the spread of pixels at each gray level for each color channel. If the uniform distribution is better, then it will be more resistant to statistical attacks. For the R, G and B channels with color image intensity between 0–255, the ideal entropy value of encrypted messages is 8; the higher the value, the more uniform the distribution. The information entropy calculation formula is as follows:(10)$$H(x)=-\sum_{i=1}^{L}P(x_{i})\log_{2}P(x_{i})$$
where $x_{i}$ is the grayscale value and *P*($x_{i}$) is the probability of the grayscale $x_{i}$. As shown in Table 4, the three-channel information entropy of the Lena ciphertext image is slightly better than the literature [42,43,44,45], indicating that the randomness of the ciphertext image based on the encryption scheme in this paper is good.

#### 4.4.4. Anti-Differential Attack Analysis

The ability to resist differential attacks is an important indicator for evaluating cryptographic algorithms. Attackers usually make minor adjustments to the plain image and then compare the differences between the cipher image produced before and after the adjustment to carry out the attack. To examine the effect of single-pixel change on the cipher image in the original image, the color map in the public image dataset USC-SIPI “Misc” was selected for the anti-differential attack analysis of the encryption scheme in this paper, and the color images of 256 × 256 and 512 × 512 correspond to the ideal pixel change rate (NPCR), and the unified average changing intensity (ACI) values are shown in Table 5 [31].

Keeping the key unchanged, randomly test each image 100 times, randomly select the value of any channel of one pixel each time and add 1, change the NPCR and UACI values of the cipher image before and after, as shown in Table 6, and average the results. In Table 6, all the pictures have passed the test, indicating that the proposed algorithm has good resistance to differential attacks and can effectively resist known plain-image attacks and selective plain-image attacks.

Keeping the key unchanged, randomly test the experimental image Lena 100 times, randomly select the value of any channel of one pixel each time and add 1, change the NPCR and UACI values of the cipher image before and after, as shown in Table 7, and average the results. Compared with the literature [42,43,44,45], the NPCR and UACI of the encryption scheme in this paper are slightly higher overall. This proves that the encryption scheme in this paper has good ability to resist differential attacks.

### 4.5. Damage Resistance Analysis

#### 4.5.1. Analysis of Noise Immunity Performance

Image noise refers to unwanted or unwanted interference information that exists in image data. In the process of image acquisition and transmission, because of the influence of image sensor material, working environment, transmission channel [48], etc., the image may be contaminated by a variety of noise, which will have a certain impact on the decryption of the terminal picture. Therefore, the ability to resist a certain intensity of salt-and-pepper-noise attacks is an indicator of the performance of image encryption algorithms. In this paper, the noise pollution in the transmission of pepper–salt noise is simulated, and different intensities of pepper–salt noise are added to the cipher image to test the noise resistance ability of the proposed algorithm.

The cipher image of the test image Lena shown in Figure 8 was decrypted by adding salt-and-pepper noise with noise densities of 0.05, 0.1, 0.15 and 0.2, respectively, and the decrypted image is shown in Figure 11. It can be seen from Figure 11 that when the salt and pepper densities are 0.05 and 0.1, respectively, the decryption algorithm in this paper can basically restore the original image. When the salt density is 0.15, the restored image is blurry, but the original image can still be recognized; when the salt density is 0.2, the restored image can still recover the general outline. The experimental results show that the greater the intensity of the noise, the deeper the impact on the image, and the worse the quality of the decrypted image, but the algorithm in this paper can still distinguish the main information of the original image from the overall visual effect, indicating that the algorithm can tolerate a certain degree of noise and have strong anti-interference ability.

#### 4.5.2. Analysis of Shear Resistance

A clipping attack is an attack method that intercepts a ciphertext image during transmission and destroys or deletes part of the data [49]. The part that is usually clipped is an area in the image that has a strong correlation between pixels, and the lost information is difficult to recover. Therefore, breaking the correlation between pixels is an indicator to measure the anti-clipping performance of image encryption algorithms. If the correlation between image pixels is strong, the cipher image after the loss of information cannot provide enough valid information, and the decryption may fail. The cipher image of the test image Lena shown in Figure 8 is cut 1/16, 1/4 and 1/2 (the pixels at the clipping position are all 0, and the clipping sample is shown in Figure 12), and the decryption of the cut cipher image is obtained as shown in Figure 13. It can be seen from Figure 13 that when the cipher image is cut by 1/16, the image can basically be decrypted and restored to the original image; when cutting 1/4, the approximate image information can still be recovered after decryption; when cutting 1/2, the restored image information can see the general outline. Experimental results show that the proposed algorithm has a certain recovery ability when subjected to shear attacks, and the proposed encryption algorithm can resist shear attacks to a certain extent.

Through a series of security experiments such as key randomness analysis, key security analysis, statistical analysis, anti-differential attack analysis and damage resistance analysis, it is proved that the encryption scheme proposed in this paper has a large key space and is a lossless color image encryption algorithm. By comparing the literature [42,43,44,45], it shows that the encryption scheme in this paper has good encryption efficiency, image information entropy, robustness and resistance to differential attacks.

## 5. Conclusions

Because images are vulnerable to external attacks in the process of network transmission and traditional image encryption algorithms have limitations such as long encryption time, insufficient entropy or poor diffusion of cipher image information when encrypting color images, this paper proposes a fast image encryption algorithm for logistics-sine-cosine mapping. A random binary number generator is used to generate 320-bit binary numbers as keys, and five sets of encrypted sequences are generated based on logistics-sine-cosine. Five sets of encryption sequences are used to scramble and spread the image pixels, and the color image is encrypted with small computing resources and high efficiency. A large number of color images of different resolutions and sizes were selected from the gallery and a series of security analysis experiments. Experimental results show that the encryption algorithm in this paper has a short encryption time, robustness and resistance to differential attacks, and is a safe and effective color image encryption algorithm, but it has certain limitations in correlation and needs to be further optimized. At the same time, although the proposed encryption algorithm is slightly better than the literature [42,43,44,45] in the time comparison experiment of encrypting Lena images, it can be seen that the encryption time of the proposed algorithm increases with the size of the encrypted image, and as the image size increases, the encryption efficiency of the proposed algorithm will gradually decrease, and further improvement is needed.

## Figures and Tables

**Figure 1 sensors-22-09929-f001:**
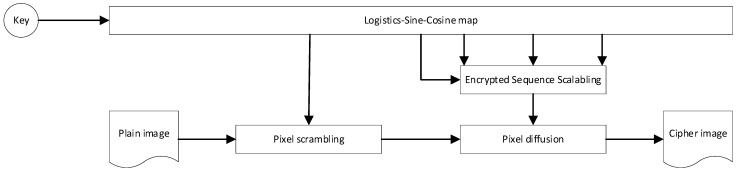
Encryption flowchart.

**Figure 2 sensors-22-09929-f002:**
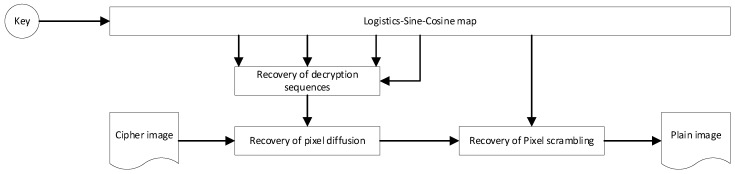
Decryption flowchart.

**Figure 3 sensors-22-09929-f003:**
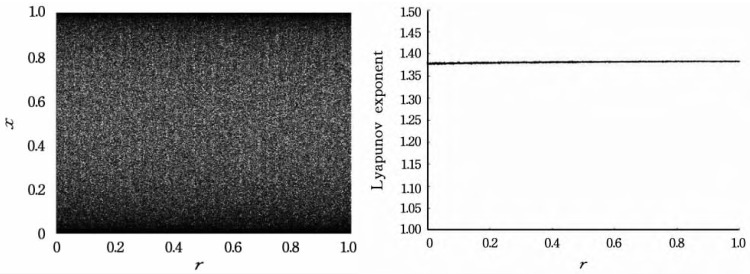
Bifurcation graph and Lyapunov exponent of logistics-sine-cosine mapping.

**Figure 4 sensors-22-09929-f004:**
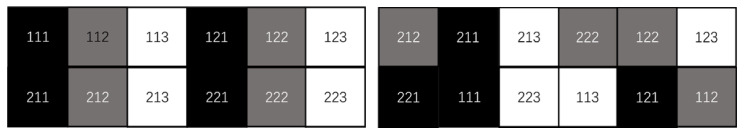
Schematic diagram of pixel position before and after scrambling.

**Figure 5 sensors-22-09929-f005:**
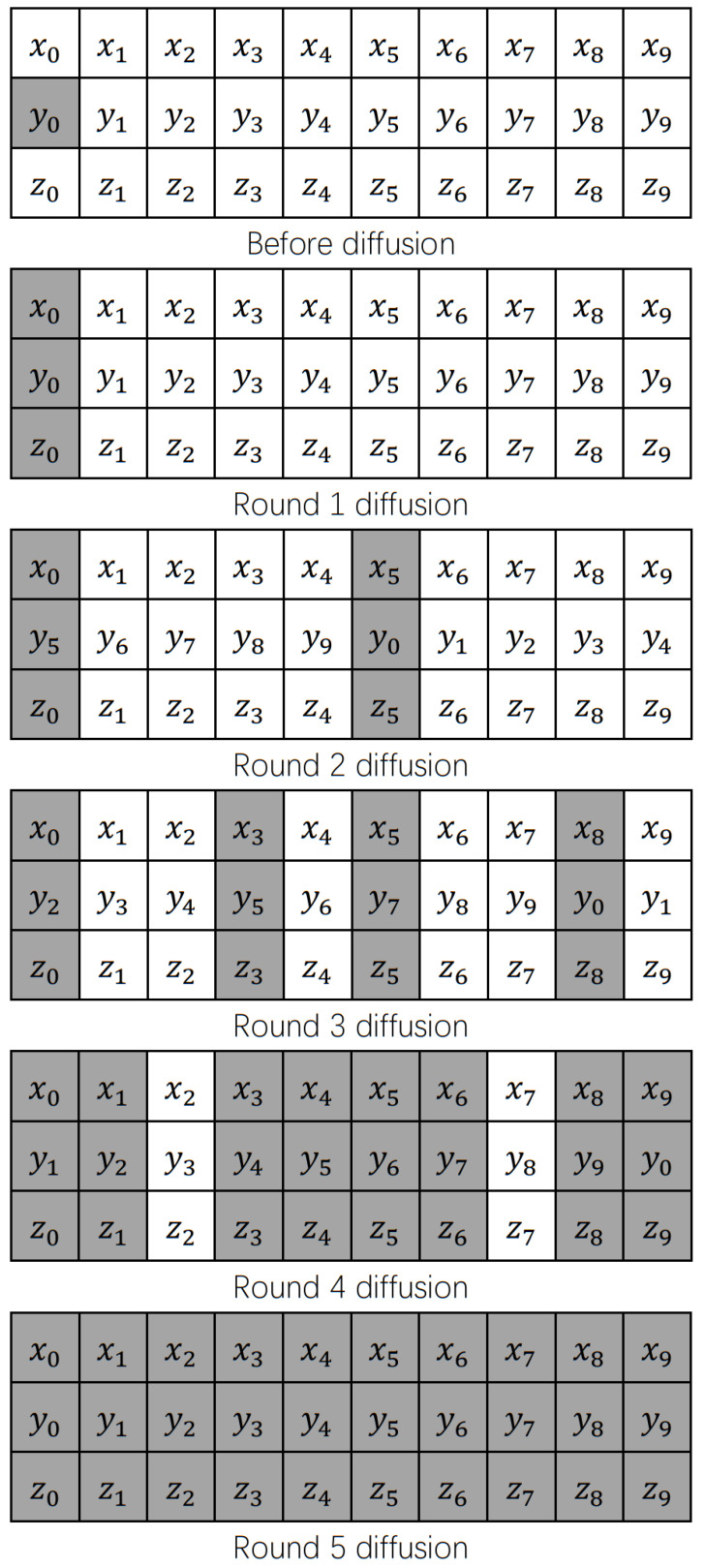
Diffusion Schematic.

**Figure 6 sensors-22-09929-f006:**
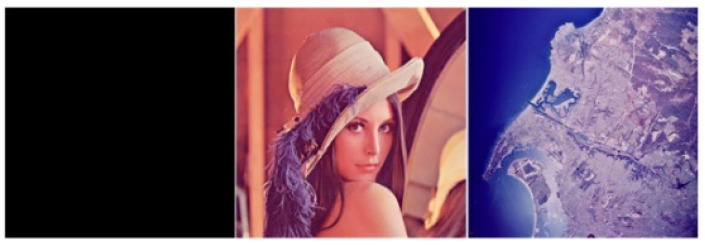
Experimental image.

**Figure 7 sensors-22-09929-f007:**
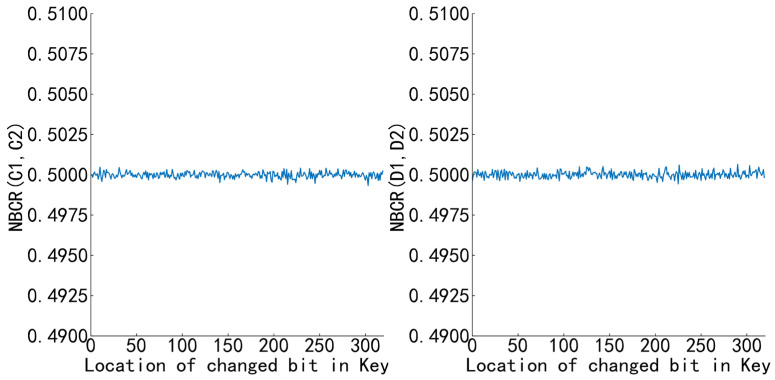
*NBCR* ($C_{1}$, $C_{2}$), *NBCR* ($D_{1},\;D_{2}$).

**Figure 8 sensors-22-09929-f008:**
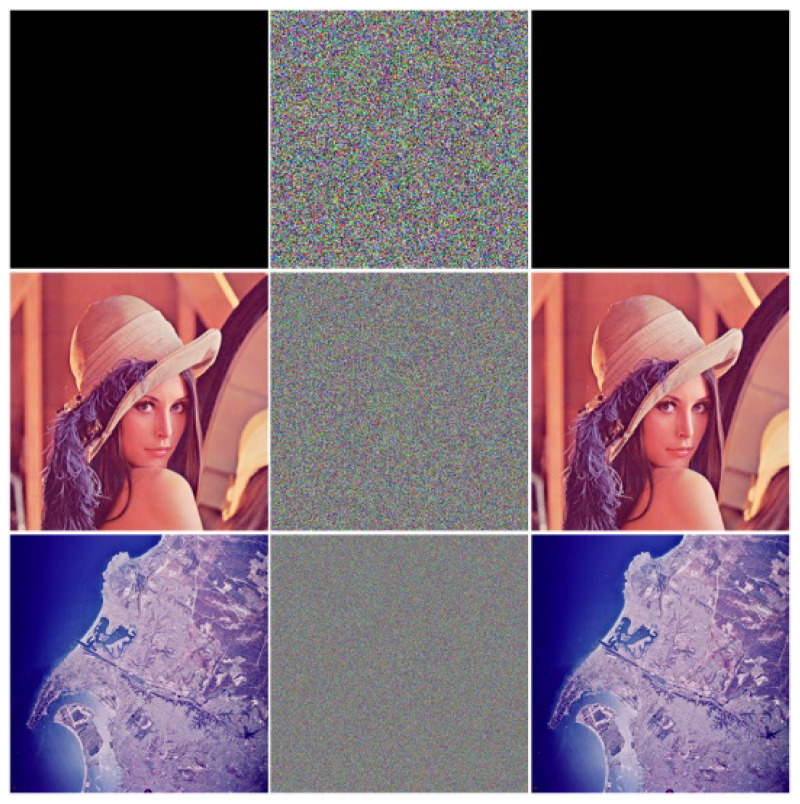
Encrypt and decode.

**Figure 9 sensors-22-09929-f009:**
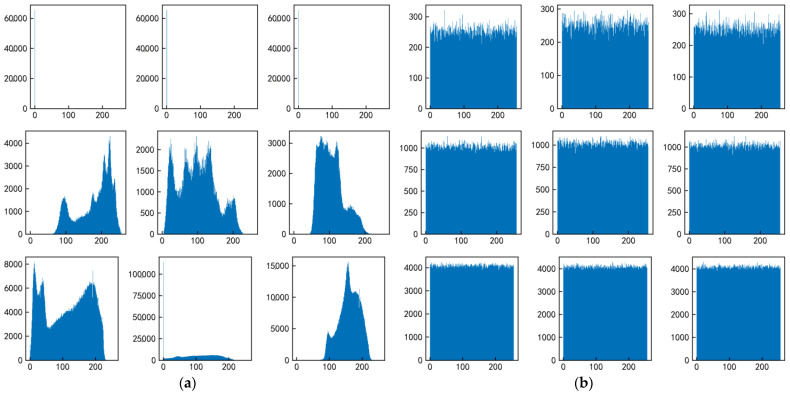
Pixel histogram of the encrypted image based on the encryption algorithm and key of the experimental image. (**a**) Histogram of plain image. (**b**) Histogram of cipher image.

**Figure 10 sensors-22-09929-f010:**
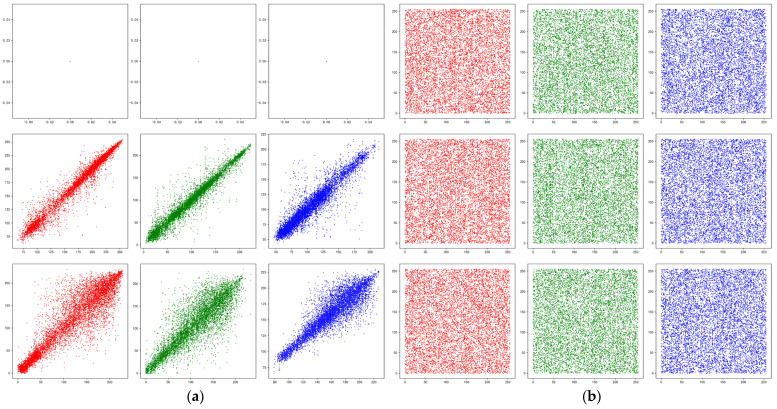
Image correlation before and after encryption. (**a**) Plain image correlation. (**b**) Cipher image correlation.

**Figure 11 sensors-22-09929-f011:**
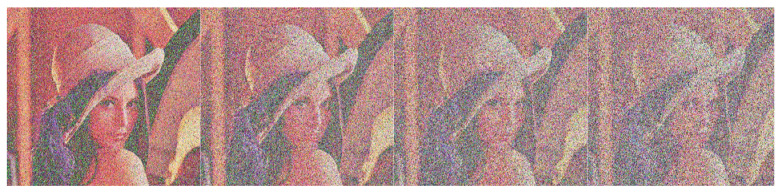
Decrypt images after different salt-and-pepper noise was added to cipher images.

**Figure 12 sensors-22-09929-f012:**
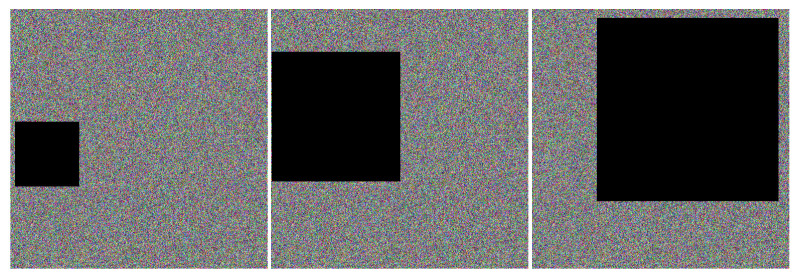
Sample encryption of secret image.

**Figure 13 sensors-22-09929-f013:**
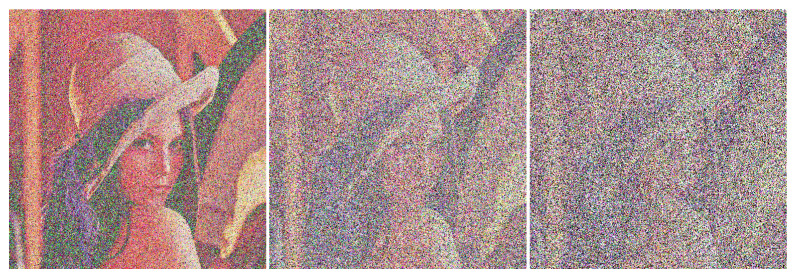
Decrypted images of cipher images cut in varying degrees.

**Table 1 sensors-22-09929-t001:** NIST statistical test results.

The Project Tested	Passed or Not
Monobit test	Pass
Frequency within block test	Pass
Runs test	Pass
Longest run ones in a block test	Pass
Binary matrix rank test	Pass
Dft test	Pass
Non-overlapping template matching test	Pass
Overlapping template matching test	Pass
Maurer’s universal test	Pass
Linear complexity test	Pass
Serial test	Pass
Approximate entropy test	Pass
Cumulative sums test	Pass
Random excursion test	Pass
Random excursion variant test	Pass

**Table 2 sensors-22-09929-t002:** Time of encrypting/decrypting.

	Test Image	Image Size	Encryption Time/s	Decryption Time/s	Time Complexity
Proposed Scheme	Black	256 × 256	0.061	0.058	O (*wh*)
	Lena	512 × 512	0.479	0.490	
	2.2.01	1024 × 1024	2.200	2.220	
Ref. [42]	Lena	512 × 512	356.34	-	O $(wh\log^{wh}$)
Ref. [43]			12.23		O (*wh*)
Ref. [45]			1.499		O (*wh*)
Logistics			0.296		O (*wh*)

**Table 3 sensors-22-09929-t003:** Three-channel correlation comparison between experimental image and cipher image.

Test Image			Horizontal	Vertical	Diagonal
Black	R	Plain	-	-	-
		Cipher	0.0007	−0.0006	−0.0043
	G	Plain	-	-	-
		Cipher	0.0008	−0.0032	0.0007
	B	Plain	-	-	-
		Cipher	−0.0004	0.0007	−0.0008
Lena	R	Plain	0.9781	0.9903	0.9697
		Cipher	−0.0031	−0.0007	−0.0009
	G	Plain	0.9717	0.9837	0.9625
		Cipher	0.0036	−0.0007	0.0008
	B	Plain	0.9352	0.9603	0.9248
		Cipher	−0.0008	−0.0009	0.0003
2.2.01	R	Plain	0.9304	0.9234	0.9083
		Cipher	0.0016	0.0009	0.0005
	G	Plain	0.9206	0.9198	0.8975
		Cipher	0.0007	−0.0010	0.0002
	B	Plain	0.9135	0.9097	0.8900
		Cipher	−0.0006	0.0005	−0.0007

**Table 4 sensors-22-09929-t004:** Comparison of information entropy of three channels before and after encryption.

	Test Image			
Proposed Scheme	Black	0.0	0.0	0.0
		7.9976	7.9978	7.9977
	Lena	6.8791	6.9262	6.9684
		7.9994	7.9994	7.9994
	2.2.01	6.5982	6.9099	6.9591
		7.9998	7.9998	7.9998
Ref. [42]	Lena	7.9992	7.9992	7.9992
Ref. [43]		7.9951	7.9965	7.9828
Ref. [44]		7.9993	7.9994	7.9994
Ref. [45]		7.9993	7.9991	7.9990

**Table 5 sensors-22-09929-t005:** Ideal values of NPCR and UACI for color images of different.

Size of Image	NPCR	UACI
256 × 256	99.5693%	33.2824~33.6447%
512 × 512	99.5893%	33.3730~33.5541%

**Table 6 sensors-22-09929-t006:** NPCR and UACI test results.

Image	Size	NPCR/%	UACI/%
4.1.01	256 × 256	99.62	33.49
4.1.02		99.61	33.52
4.1.03		99.62	33.50
4.1.04		99.62	33.48
4.1.05		99.61	33.48
4.1.06		99.61	33.50
4.1.07		99.62	33.55
4.1.08		99.61	33.50
4.2.01	512 × 512	99.61	33.48
4.2.03		99.61	33.47
4.2.05		99.61	33.48
4.2.06		99.61	33.47
4.2.07		99.61	33.49
house		99.61	33.48

**Table 7 sensors-22-09929-t007:** Comparison of NPCR and UACI for the Lena by different encryption algorithms.

		R	G	B
Proposed Scheme	NPCR/%	99.62	99.62	99.62
	UACI/%	33.49	33.48	33.49
Ref. [42]	NPCR/%	99.61	99.63	99.61
	UACI/%	33.45	33.46	33.45
Ref. [43]	NPCR/%	99.61	99.58	99.62
	UACI/%	33.36	33.49	33.5
Ref. [44]	NPCR/%	99.61	99.61	99.61
	UACI/%	33.46	33.48	33.45
Ref. [45]	NPCR/%	99.60	99.64	99.61
	UACI/%	33.43	33.48	33.55

## Data Availability

Not applicable.

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
