# Peer review of "Fast Image Encryption Algorithm for Logistics-Sine-Cosine Mapping"

_sensors, 2022, doi:10.3390/s22249929_

Round 1

Reviewer 1 Report

The paper is well-written and contains all important test results. It is need some revisions:

1- The algorithmic complexity of the classical system and the proposed fast system must be compared. Also, the complexity of the compared methods must be given in the time results table.

2- The literature contribution of the proposed paper must be given clearly.

3- The main problem is about the references and related works. The manuscript contains 37 references and 21 of them [7-27] are given in a sentence. It is not an acceptable approach. 

4-The compared methods did not discuss enough and were only given as [34-37] in the manuscript. They must be  

5-Related work is weak and only contains 4 references. It must be enriched with a detailed discussion.

Author Response

Thank you very much for taking the time to review my paper, I have supplemented the shortcomings you pointed out for me and explained them in the annexes, and I very much look forward to your comments on my revised draft.

Author Response

(The authors gave the same response as above.)

Round 2

Reviewer 1 Report

All my findings and concerns were solved.

Reviewer 2 Report

The authors have considerably improved the quality of the proposed work. In this form, this work may be considered for publication.